# Preparation and Evaluation of Auxiliary Permeable Microneedle Patch Composed of Polyvinyl Alcohol and Eudragit NM30D Aqueous Dispersion

**DOI:** 10.3390/pharmaceutics15072007

**Published:** 2023-07-22

**Authors:** Mengzhen Xing, Yuning Ma, Xiaocen Wei, Chen Chen, Xueli Peng, Yuxia Ma, Bingwen Liang, Yunhua Gao, Jibiao Wu

**Affiliations:** 1Key Laboratory of New Material Research Institute, Institute of Pharmacy, Shandong University of Traditional Chinese Medicine, Jinan 250355, China; mengzhen@mail.ipc.ac.cn (M.X.); 60210001@sdutcm.edu.cn (Y.M.); 60050072@sdutcm.edu.cn (X.W.); 21129008@zju.edu.cn (C.C.);; 2College of Traditional Chinese Medicine, Shandong University of Traditional Chinese Medicine, Jinan 250355, China; 3Qingdao Academy of Chinese Medical Sciences, Shandong University of Traditional Chinese Medicine, Qingdao 266112, China; 2022111558@sdutcm.edu.cn; 4Department of Acupuncture-Moxibustion and Tuina, Shandong University of Traditional Chinese Medicine, Jinan 250355, China; 60050012@sdutcm.edu.cn; 5Key Laboratory of Photochemical Conversion and Optoelectronic Materials, Technical Institute of Physics and Chemistry of Chinese Academy of Sciences, Beijing 100190, China; 6Beijing CAS Microneedle Technology Ltd., Beijing 102609, China

**Keywords:** auxiliary permeation microneedles, polyvinyl alcohol, Eudragit NM30D, skin permeability, transdermal drug delivery

## Abstract

Poor transdermal permeability limits the possibility of most drug delivery through the skin. Auxiliary permeable microneedles (AP-MNs) with a three-dimensional network structure can effectively break the skin stratum corneum barrier and assist in the transdermal delivery of active ingredients. Herein, we propose a simple method for preparing AP-MNs using polyvinyl alcohol and Eudragit NM30D for the first time. To optimize the formulation of microneedles, the characteristics of swelling properties, skin insertion, solution viscosity, and needle integrity were systematically examined. Additionally, the morphology, mechanical strength, formation mechanism, skin permeability, swelling performance, biocompatibility, and in vitro transdermal drug delivery of AP-MNs were evaluated. The results indicated that the microneedles exhibited excellent mechanical-strength and hydrogel-forming properties after swelling. Further, it proved that a continuous and unblockable network channel was created based on physical entanglement and encapsulation of two materials. The 24 h cumulative permeation of acidic and alkaline model drugs, azelaic acid and matrine, were 51.73 ± 2.61% and 54.02 ± 2.85%, respectively, significantly enhancing the transdermal permeability of the two drugs. In summary, the novel auxiliary permeable microneedles prepared through a simple blending route of two materials was a promising and valuable way to improve drug permeation efficiency.

## 1. Introduction

In the past 20 years, microneedles (MNs) have been extensively studied in transdermal drug delivery because of their high efficiency, safety, and friendliness [1,2,3]. To meet the clinical medication needs of different diseases, drugs, and drug release rates, MNs, including solid microneedles (SMNs), hollow microneedles (HMNs), coating microneedles (CMNs), dissolving microneedles (DMNs), hydrogel-forming microneedles (HF-MNs), and frozen microneedles (FMNs) had been developed, successively [4,5,6]. Herein, depending on whether or not the drug is directly loaded onto MNs, the MNs can be classified as needle-drug integrated microneedles or auxiliary permeation microneedles (AP-MNs), respectively. Compared to integrated MNs, AP-MNs have advantages in clinical applications as they can be used alone as medical devices. Further, AP-MNs are divided into SMNs and HF-MNs. Among them, representatives of SMNs, such as monocrystalline silicon MNs and metal MNs, have been widely used in the clinical and medical aesthetics fields in the form of roller MNs and nano-electric MNs [7,8], while the HF-MNs, prepared based on the physical cross-linking, electrostatic interaction, and covalent chemical cross-linking of polymer chains, were still in the preclinical research stage [9,10,11]. After insertion into the skin and absorption tissue fluid, HF-MNs form a continuous and unblockable network channel, allowing the diffusion and release of drugs applied onto the substrate of MNs [12]. For the AP-MNs, the difference between HF-MNs and SMNs is that the former remains on the skin after use, and the drug-loaded reservoir is directly superimposed on the base of MNs, while the latter are immediately removed once treating the skin, then separately administered with a medicated formulation. Therefore, the use of HF-MNs can achieve more efficient and long-term drug delivery by avoiding the problem of rapid healing of skin microchannels, and it has become a hotspot in the field of AP-MNs in recent years [13,14].

In 2012, the Donnelly group first reported AP-MNs prepared with hydrogel materials for enhanced transdermal drug delivery [15]. In detail, it was obtained through chemical crosslinking based on the esterification reaction of poly (methyl vinyl ether/maleic anhydride) and poly (ethylene glycol) at high temperature (80 °C) for 24 h [16,17]. Likewise, polyvinyl alcohol (PVA), a biodegradable polymer frequently applied in biomedical systems, was also commonly used to prepare AP-MNs due to its polyhydroxy groups and phase transformation characteristics. Generally, the preparation process involves the usage of crosslinking agents or repeated freeze–thaw cycles [18,19,20]. Overall, the manufacturing processes of AP-MNs mentioned above all had defects, such as harsh reaction conditions, time-consuming, and inconvenient industrial production. Developing novel AP-MNs with friendly fabrication conditions and easy mass production was necessary to improve the transdermal permeability of clinical drugs with poor delivery efficiency.

In 2018, our team first proposed using pharmaceutical acrylic resin (Eudragit) for MN preparation and developed HF-MNs loaded with granisetron for sustainable drug release [21]. On this basis, research has been conducted on using different types and functions of Eudragit to fabricate AP-MNs. Fortunately, it was found that the lotion polymerization products of Eudragit aqueous dispersion may be a suitable material due to its excellent water solubility, film-forming property, and water vapor barrier after forming the dry film [22]. The aqueous dispersion of Eudragit is a water-based system composed of solid or semi-solid spherical or quasi-spherical particles of 10–100 nm dispersed in water, containing Eudragit NM30D, Eudragit NE30D, Eudragit FS30D, etc. [23]. Mainly, Eudragit NM30D is a neutral acrylic resin aqueous dispersion consisting of the copolymer of ethyl acrylate and methyl methacrylate with a molar ratio of 2:1. As a skeleton material, Eudragit NM30D has wide applications in the field of drug coating and prolonging or controlling release, such as solid dispersion, enteric-coated formulations, etc. [24,25]. In addition, Eudragit NM30D exhibits film-forming properties. During the film-forming process, the latex particles form a stacking layer. As the water evaporates, the surface tension increases, causing the latex particles to tightly aggregate, forming a dense and excellent water-permeable film. Unfortunately, it is too soft to show mechanical strength. Therefore, it must be used with other materials for MN preparation.

PVA with a high alcoholysis degree (>98%, H-PVA) dissolves in hot water above 90 °C. Due to its high crystallinity, it has good mechanical properties, making it commonly used to develop polymeric microneedles [26,27,28]. Unlike the low-alcoholysis PVA (L-PVA) MNs that quickly dissolve after insertion into the skin [29,30], the H-PVA molecules contain more hydroxyl groups. Once the H-PVA aqueous solution is dried and film-formed, more intramolecular and intermolecular hydrogen bonds are generated, which increases the cohesive force of the PVA molecules. H-PVA MNs will not completely dissolve after water immersion but form an irregular hydrogel cluster. Our other study found that H-PVA MNs created a porous network structure after absorbing phosphate (PBS) buffer, which is suitable for promoting drug permeation channels (not published). However, unlike PVA MNs after freeze–thaw treatment or chemical reaction cross-linking, the MNs obtained by hydrogen bond interaction alone cannot guarantee the integrity of MN arrays after swelling, making it not meet the requirements of AP-MNs. 

In this study, to develop a simple and feasible preparation process of AP-MNs for enhancing transdermal drug permeation, Eudragit NM30D and PVA were selected as the optimal matrix materials. The AP-MNs were successfully designed through a simple blending method at a specific proportion. Afterward, the morphologies of MNs before and after swelling, skin insertion property, swelling ratio, and cell safety were characterized and measured. Moreover, two model drugs, azelaic acid (AZA) and matrine (MAT), were used to evaluate the permeability of AP-MNs according to the typical characteristics of the two drugs. On the one hand, they are acidic and alkaline drugs, respectively, and in terms of solubility, AZA is slightly soluble in water, while MAT is easily soluble. On the other hand, both ingredients have significant therapeutic effects on skin diseases, but their transdermal permeability is not satisfactory [31]. 

## 2. Materials and Methods

### 2.1. Materials 

Polyvinyl alcohol (PVA) (alcoholysis degree/mol: 98.0–99.8%) was purchased from Sinopharm Chemical Reagent Co., Ltd. (Shanghai, China). Medicinal excipient-grade Eudragit NM30D was freely given by Evonik Specialty Chemicals (Shanghai, China) Co., Ltd. Azelaic acid (AZA) (purity: 99.56%) was purchased from Senxuan Pharm Co., Ltd. (Taizhou, China). Matrine (MAT) (purity: 98.00%) was obtained from Zijinhua Pharm Co., Ltd. (Wuzhong, China). Dulbecco’s modified eagle medium (DMEM) was acquired from Hyclone (Logan, UT, USA). Fetal bovine serum (FBS), penicillin-streptomycin (P/S), and 0.25% Trypsin were obtained from Gibco (Grand Island, NY, USA). Sodium 1-heptanesulfonate (Grade: HPLC) was purchased from Sysw Biotech Co., Ltd. (Qingdao, China). Acetonitrile was bought from Thermo Fisher Scientific (Waltham, MA, USA), while phosphoric acid was obtained from Aladdin (Shanghai, China). Cell Counting Kit-8 was purchased from Dojindo (Kumamoto, Japan). An acridine orange/ethidium bromide (AO/EB) dual fluorescence staining kit was supplied by Yuanye Biotech Co., Ltd. (Shanghai, China). Porcine cadaver skin was purchased from Kaikai Tech Co., Ltd. (Shanghai, China). Fibroblasts (L929) and human immortalized keratinocytes (HaCaT) were kindly supplied as a present from Professor Zhongwei Niu, Technical Institute of Physics and Chemistry (Beijing, China). All other chemicals and materials used in this study were of analytical grade.

### 2.2. Fabrication of AP-MNs

First, PVA solution was dissolved at 90 °C, and then the PVA solution was naturally cooled to room temperature. The aqueous dispersion of Eudragit NM30D was added into the PVA solution, then stirred, mixed evenly, and centrifuged at 5000 rpm for 10 min to obtain the microneedle preparation solution. The self-made polydimethylsiloxane (PDMS) microneedle mold (500 μm height, 500 μm intervals, 144 tips) was used to fabricate AP-MNs under vacuum conditions, according to our previous study [32]. In detail, the quantitative microneedle solution was poured into the PDMS mold units through a liquid injector and spread evenly with a glass rod. Afterward, it was filled into each pinhole under vacuum conditions. Then, it dried naturally at room temperature until the microneedles formed. Finally, the microneedles were stored in a drying cabinet at 25 °C and 15% relative humidity for backup.

### 2.3. Formulation Optimization of PVA and Eudragit NM30D

To acquire AP-MNs with a more straightforward process, H-PVA and hydrophobic Eudragit NM30D were selected as the microneedles’ matrix materials. First, PVA solutions of 10%, 12%, and 15% (*w*/*v*), respectively, were prepared. The viscosity values were measured via a viscometer (DV2T, BROOKFIELD, Worcester, MA, USA) to screen the optimal content of PVA. Next, a series of microneedle solutions with different PVA and Eudragit NM30D ratios were fabricated. Different proportions of each formulation are shown in Appendix A. Each microneedle patch was immersed in 3 mL PBS (0.01 M, pH = 7.4) solutions and placed in a 37 °C incubator for 24 h. They were subsequently observed and the integrity was recorded after swelling. Simultaneously, the skin insertion property of all microneedles was tested according to the trypan blue staining method. In addition, a quantitative evaluation of the swelling ratio was conducted on microneedles with swelling ability. Specifically, the microneedles were dehydrated, weighed, and recorded as W_0_. After that, they were immersed in a vial containing PBS preheated to 37 °C and maintained at constant temperature for eight hours. Next, all microneedles were removed from the vial, the excess PBS buffer on their surfaces was wiped off, and they were weighed again as W_1_. The percentage of swelling was calculated through the following Equation (1).
(1)$$Swelling\;ratio\;\left(\%\right)=\frac{W_{1}}{W_{1}-W_{0}}\times 100\%$$

Based on these, a microneedle formulation consisting of 12% PVA and 4% Eudragit NM30D, with moderate viscosity, swelling behavior, and skin insertion performance was determined and further optimized. In detail, microneedle prescriptions were prepared and tested for swelling ratio, including 12% PVA + 5% Eudragit NM30D and 12% PVA + 6% Eudragit NM30D. Moreover, to investigate the integrity of microneedles’ tips after swelling, the morphology of each microneedle formulation was measured after applying it on isolated piglet skin for 8 h. First, the adhesive hydrogel backings were stuck to the base layer of microneedles. Then, they were placed on the stratum corneum of porcine skin. The microneedle patches were pressed for 20 s with the assistance of a self-made applicator for 8 h. Subsequently, the microneedles were removed and their morphological changes were observed through a fluorescence microscope (BX51, Olympus, Hachioji, Japan).

### 2.4. Determination of Matrix Solution Addition

The preferred microneedle solution composed of 12% PVA + 6% Eudragit NM30D was prepared with a liquid addition of 20, 30, and 40 μL. Afterward, each microneedle was soaked in PBS buffer for 1, 12, and 24 h, and then the swelling ratio was measured according to Equation (1). Meanwhile, the swollen microneedles were put back into the vials, dried in an oven at 40 °C to constant weight, weighed as W_2_, and the gel fraction of AP-MNs was calculated based on Equation (2).
(2)$$Gel\;fraction\left(\%\right)=\frac{W_{2}}{W_{1}}\times 100\%$$

### 2.5. Fourier Transform Infrared (FTIR) Spectroscopy Analysis

FTIR measurements were carried out using an Excalibur 3100 FTIR spectrometer (Varian, Palo Alto, CA, USA) equipped with an attenuated total reflectance (ATR) attachment. First, PVA, Eudragit NM30D, and PVA + Eudragit NM30D microneedles were prepared and tested, respectively. The spectrum was read over 32 repeated scans at 4 cm^−1^ resolutions, and the wavenumber range was acquired from 600 to 4000 cm^−1^. 

### 2.6. Morphological Characterization

Visualization of microneedles prepared with PVA, Eudragit NM30D, as well as PVA + Eudragit NM30D was performed utilizing a stereomicroscope (SMP1000, Nikon, Tokyo, Japan) fluorescence microscope (BX51, Olympus, Hachioji, Japan), respectively. Moreover, scanning electron microscopy (SEM) (3 kV, MIRA LSM, TESCAN, Brno, Czech Republic) was used to analyze the internal morphologies of three microneedles before and after swelling in PBS buffer for 24 h.

### 2.7. Skin Permeability

First, skin insertion measurement was applied on three kinds of microneedles. In detail, each microneedle was implanted into porcine skin ex vivo for 20 s, assisted with a custom-made applicator (20 N/cm^2^). Afterward, the skin was stained with 4 mg/mL trypan blue dye for 30 min and observed through the microscope in brightfield. Second, the skin permeability of AP-MNs was assessed via a laser confocal microscope (LSM880, ZEISS, Oberkochen, Germany). A hollow 3M adhesive patch was attached to the microneedles’ substrate layer, allowing the drug solution to be directly delivered into the skin through AP-MNs. The 1% sodium fluorescein (SF) solution was used as a model drug. Simultaneously, the same dose of SF solution was pipetted into the stratum corneum skin as a reference. 

### 2.8. Cell Activity and Biocompatibility

Based on our previous study, L929 and HaCaT cells were determined to evaluate the cytotoxicity of AP-MNs. After being disinfected by exposure to ultraviolet light, one-piece AP-MNs were soaked in 1 mL DMEM medium. Subsequently, the supernatants were obtained and diluted into 100%, 50%, and 25% concentrations incorporated with 10% FBS and 1% P/S. Meanwhile, two cells were seeded at a density of 5 × 10^3^ cells and incubated for 24 h. Then the medium was replaced with the AP-MNs extracted. After incubation for 24 h, the CCK-8 kit was used to detect cell viability according to a manufacturer’s protocol [33]. Besides, the live/dead situation of two cells cultured with AP-MN-extracted medium was assessed using AO/EB dual fluorescence staining kit, according to our previous study [34].

### 2.9. In Vitro Swelling Properties

The swelling performance of AP-MNs was investigated using the same method previously explained in Section 2.3. AP-MNs were weighed and placed in a glass containing 3 mL PBS buffer at 37 °C. At regular intervals (0.5, 1, 2, 3, 6, 12, and 24 h), each AP-MN was removed and the excess buffer was wiped off using filter paper. The swelling ratio was calculated by using Equation (1). To investigate the morphology changes of AP-MNs during the swelling process, 5% agar gel was applied to simulate the ex vivo skin, and the appearance and height of microneedles after being inserted for a specific time (0, 1, 2, 3, and 4 h) were measured through the stereomicroscope (SMP1000, Nikon, Tokyo, Japan) and fluorescence microscope (BX51, Olympus, Hachioji, Japan).

### 2.10. In Vitro Percutaneous Permeability Test for Acidic and Alkaline Drugs of AP-MNs

To verify the permeability capability of AP-MNs, azelaic acid (AZA) and matrine (MAT), an acidic drug slightly soluble in water and an alkaline drug easily soluble in water, respectively, were chosen as model drugs to conduct in vitro transdermal penetration experiments of two drug aqueous solutions. First, the thawed piglet skin was cut into pieces of uniform size and thickness (600 μm). Subsequently, it was soaked in PBS buffer to equilibrate for 1 h at 37 °C. The AP-MN patches attached with hollow backing adhesive were inserted into the stratum corneum side of the skin via a 20 N applicator. Next, the skin was fixed between the donor and receptor compartments of the Franz cell, and PBS buffer (0.01 M, pH = 7.4) was used as receiving solution. Then, a 100 μL drug solution consisting of 1% AZA and 1.3% MAT was added into the substrate layer of the AP-MNs. Samples were collected from the receptor compartment and supplied with the same volume of fresh PBS buffer at 0.5, 1, 2, 3, 4, 6, 8, 10, 16, and 24 h. The process was performed via modified Franz diffusion cells (912-SCT-S, Logan Instruments Corporation, Somerset, NJ, USA), and the magnetron at the bottom of the receptor cell was set to perform magnetic stirring at 280 rpm. Simultaneously, the same dosage and concentration of AZA and MAT solution were directly poured into the stratum corneum skin, which was regarded as a control group. Moreover, the contents of AZA and MAT in each sample were determined via the HPLC method, as described in our previous study [31]. 

### 2.11. Statistical Analysis

Microsoft Excel 2010 software was used to analyze statistical data. Origin 2018 software was used to plot graphs. The values of * *p* < 0.05 and ** *p* < 0.01 were considered statistically significant.

## 3. Results and Discussion

### 3.1. Determination of AP-MN Formulation

Based on the usage characteristics of AP-MNs, the microneedle formulation was determined according to swelling properties, skin insertion, solution viscosity, swelling ratio, and the integrity of needles’ tips taken out after application to the skin for 8 h. As shown in Appendix A, the microneedles were completely dissolved or lost their morphological integrity after being immersed in PBS buffer for 24 h, when the Eudragit NM30D content in the microneedles’ formulation was less than 4% (*w*/*v*). On the contrary, when the Eudragit NM30D content was above 4%, the morphology of the microneedle was intact after mixing with different content of PVA due to the excellent film-forming and water resistance of Eudragit aqueous dispersion [35]. Simultaneously, the mechanical strength of each microneedle prescription was investigated through the skin insertion experiment, and the results were presented in Appendix A. Specifically, PVA microneedles had a good puncture effect without Eudragit NM30D added due to the crystallinity of PVA [36]. Nevertheless, with the addition of Eudragit, the mechanical strength of the microneedles was decreased, manifested by the incomplete pinhole array after puncturing the skin with microneedles composed of 10% PVA and 4% Eudragit NM30D. Moreover, the viscosity results of 10%, 12%, and 15% PVA aqueous solutions were 2994 ± 60, 9940 ± 200, and 50,900 ± 1000 cp, respectively (Figure 1A). Compared to the first two, the viscosity of the 15% PVA solution was too high to flow, as shown in Appendix A. Simultaneously, the excessive-viscosity solution cannot be easily inhaled or dropped via the liquid injector, decreasing the accuracy of the addition amount. Accordingly, the PVA solutions with concentrations of 10% and 12% were preferred to use.

In Figure 1B, the swelling ratio of microneedles increased with the increase of PVA content in the formulation when the content of Eudragit NM30D was 4%. Among them, the swelling ratio of microneedles composed of 15% PVA + 4% Eudragit NM30D presented a significant difference compared with those composed of 10% PVA + 4% Eudragit NM30D microneedles. In contrast, a decrease in swelling ratio was observed with an increase in Eudragit NM30D content in microneedles (Figure 1C), although there was no apparent difference between these data (*p* > 0.05). These results were consistent with our speculation. Owing to the hydrophilicity of PVA, the higher content of PVA, the stronger its water absorption and the greater the swelling ratio of the microneedle. On the contrary, the hydrophobicity of Eudragit NM30D makes it difficult for microneedles with higher Eudragit content to absorb water and swell, resulting in a decrease in swelling ratio. Further, the integrity of microneedles’ tips after applying on ex vivo skin for 8 h was tested to optimize the composition of AP-MNs. The results are exhibited in Figure 1D. Consistent with the results in Figure 1C, with the increase of Eudragit NM30D content, the swelling property of microneedles decreased, and the integrity of needles’ tips increased. Primarily, the microneedles consisting of 12% PVA + 6% Eudragit NM30D presented a more complete microneedle morphology after swelling for 8 h, so this was considered the best formulation for AP-MNs. On account of the appropriate proportion between hydrophilic PVA and hydrophobic Eudragit NM30D, it formed a three-dimensional network structure. Furthermore, the water-resistant framework created by Eudragit aqueous dispersion after film formation blocked water vapor penetration [35]. In comparison, the unbounded hydroxyl groups in PVA molecules easily formed weak hydrogen bonds with water molecules, making them swell into hydrogel in water. When the two materials were mixed appropriately to prepare microneedles, the obtained AP-MNs quickly absorbed water and swelled to form a three-dimensional network while maintaining the shape of complete needles. Therefore, the hydrogel network channel can realize the lasting delivery of drugs and can be completely removed from the skin surface after use.

### 3.2. Determination of Addition Amount of MN Solution

As displayed in Figure 2A, the swelling ratio of AP-MNs was related to the composition and proportion between PVA and Eudragit NM30D and the addition amount of microneedle solution via a micro-mold casting method. The swelling ratios of microneedles prepared with an addition amount of 20 μL per well were 442.31 ± 9.47%, 343.11 ± 13.83% (*p* < 0.01), and 339.59 ± 19.28% (*p* < 0.01), respectively, after immersion in PBS buffer for 1, 12, and 24 h. This revealed that the microneedles were unstable after swelling and maybe degraded when applied to the skin for a long time. On the contrary, those ratios were 428.13 ± 21.51%, 407.61 ± 15.12% (*p* > 0.05), and 416.57 ± 12.21% (*p* > 0.05), respectively, with an addition amount of MN solution of 30 μL per well. Further, the swelling ratios at the addition amount of 40 μL were 390.23 ± 9.75%, 385.71 ± 9.40% (*p* > 0.05), and 388.06 ± 7.01% (*p* > 0.05) after swelling for 1, 12, and 24 h, respectively. The experimental results showed that the stability of the microneedle formulations was positively correlated with the addition amount of MN solution. There was no significant difference in the swelling ratio measured after immersion in PBS buffer for 1, 12, and 24 h between microneedles prepared with an addition amount that was not less than 30 μL. Similarly, it was also found in the gel fraction experiment of three microneedles (Figure 2B). Compared with microneedles at an addition amount of 20 μL, the gel fractions of microneedles of 30 and 40 μL addition amounts were higher and more stable. In addition, considering that the volume of the microneedles’ tips was constant, increasing the amount of microneedle solution increased the thickness of the microneedles’ basal layers, which led to decreased drug permeation efficiency. Therefore, it was determined that the addition amount of MN solution was 30 μL per well.

### 3.3. Three-Dimensional Network Structure Characterization of AP-MNs

FTIR was employed to examine the interaction between PVA and Eudragit NM30D in forming AP-MNs. The alkyl (C-H) stretching vibration peaks of PVA and Eudragit NM30D were 2971 and 2969 cm^−1^, respectively. After mixing, the C-H stretching vibration peak shifted to a lower wavenumber of 2940 cm^−1^, indicating a weak hydrophobic interaction between the two materials. Compared with the C-O transmission peak of PVA, the peak of PVA in the AP-MNs showed no change. Besides, the characteristic peaks (C=O) of Eudragit NM30D before and after being mixed with PVA were both 1727 cm^−1^ (Figure 3), revealing that no interaction was observed between PVA and Eudragit NM30D [37]. The study revealed that AP-MN preparation does not involve inter-molecular chemical interaction. 

The macroscopic and microscopic morphologies before and after the swelling of AP-MNs were further characterized. As presented in Figure 4A, the microneedles prepared by separate PVA and Eudragit NM30D were transparent after molding, while the AP-MNs were white. The microneedles made of PVA were found to have a partial fracture at the needles’ tips under a bright field microscope, which may be due to the high crystallinity of PVA, resulting in poor flexibility [38] (Figure 4(B-1)). Meanwhile, the microneedles made by Eudragit NM30D were too soft to insert into the ex vivo skin, and the interior of the microneedles was hollow under a microscope (Figure 4(B-2)). After blending the two materials, the properties of the microneedles were optimized in all aspects, characterized by sharp needle tip, solid needle body, good flexibility, and sufficient mechanical strength to penetrate the skin stratum corneum (Figure 4(B-3)). 

Furthermore, the micro-structure of microneedles before and after swelling was recorded via SEM in Figure 4C,D. Before swelling, the surface of PVA MNs was rough, and a small amount of pore distribution could be observed under the SEM (Figure 4(C-1)). On the contrary, the surface of Eudragit NM30D MNs was extremely dense (Figure 4(C-2)). The MNs prepared by blending two materials combined the characteristics of both materials, reflected in the porous and rough surface of AP-MNs with dense and compact structures (Figure 4(C-3)). As is well known, the three-dimensional network structure of HF-MNs is the foundation for promoting drug penetration [12,39]. For PVA MNs, the hydrophilicity of hydroxyl groups in PVA molecules makes it easy to swell in water. Because of the strong hydrogen bond interaction formed by part hydroxyl groups, it will not be completely dissolved in water but will create a deformed hydrogel. As Figure 4(D-1) shows, irregular pores appeared inside the microneedles forming filaments and clumps. For Eudragit NM30D MNs, there was no apparent change in the internal morphology before and after swelling due to the water resistance of Eudragit aqueous dispersion (Figure 4(D-2)). Compared with them, the MNs composed of PVA and Eudragit NM30D swelled to produce uniform, continuous, and unobstructed channels with a 3D network structure, which is the basis for evaluating the permeability performance of AP-MNs (Figure 4(D-3)) [15]. 

In summary, we speculate that the formation mechanism of AP-MNs based on the combination of PVA and Eudragit NM30D is due to the blending and encapsulation effect between the two materials. Eudragit NM30D can be mixed with aqueous solution at any ratio before film formation to disperse with PVA molecules in the solution state evenly. Once dried, Eudragit NM30D loses its hydrophilicity and hardly absorbs water in PBS buffer solution, while PVA can quickly absorb water and swell. Therefore, the dense skeleton of Eudragit NM30D can effectively regulate the swelling degree of PVA, thus forming a hydrogel network system that can promote drug penetration. In comparison with the preparation processes of AP-MNs reported previously, such as repeated freeze–thaw, high temperature, UV irradiation, etc. [40,41,42], the formulation composed of PVA and Eudragit NM30D based on simple blending has apparent advantages in the preparation process and batch implementation.

### 3.4. Permeability and Safety of AP-MNs

The permeability of microneedles prepared with PVA, Eudragit NM30D, and PVA + Eudragit NM30D was evaluated. In Figure 5A, ex vivo skin insertion results showed that both PVA MNs and AP-MNs had good mechanical strength and could effectively break the skin stratum corneum barrier. However, there was no skin puncture ability on Eudragit NM30D MNs, consistent with the results in Appendix A. Next, SF, a water-soluble dye, was used as a model drug to study the transdermal delivery of medications under the assistance of AP-MNs [43]. The schematic diagram was shown in Figure 5B. With the adhesive force of the hydrogel backing, AP-MNs were effectively fixed and inserted into the skin. The microneedles absorbed the interstitial fluid and swelled, forming three-dimensional network microchannels in AP-MNs. Then, the reservoir of the drug solution was poured into the base layer of AP-MNs. Under the protection of the surrounding hydrogel backing, it could not cause the diffusion and overflow of the drug solution. Therefore, the small-molecule drugs stored in the reservoir were passively diffused and successfully delivered to the intradermal tissue through AP-MNs’ microchannels. 

As exhibited in Figure 5C, due to the barrier effect of the stratum corneum, SF was almost unable to pass through the intact skin, which was also a true reflection of most skincare products after application. In other words, many active ingredients cannot be effectively delivered to the epidermis or dermis [44,45,46]. Fortunately, with the assistance of AP-MNs, SF can efficiently penetrate the skin’s dermis with a permeability depth of approximately 200 μm after applying for 30 min (Figure 5D) and even be transported through capillaries to achieve systemic drug delivery. This result was consistent with our previous study [31]. Owing to the elasticity of the skin and the cushioning effect of subcutaneous fat, the exact penetration depth was generally shorter than the actual height of the microneedles [47]. Moreover, the depth of 200 μm under the stratum corneum is the junction site of the basal layer and the dermis in human skin. There is no nerve and blood vessel distribution, which can effectively prevent pain and bleeding caused by acupuncture [31,48]. Meanwhile, drugs penetrating to here will be passively diffused and transported to capillaries in the dermis and delivered throughout the body via the microcirculation system.

To assess the safety of AP-MNs, cytotoxicity and cell viability staining tests were conducted using L929 and HaCaT cells. Compared to cells incubated with normal mediums, the AP-MN extract showed excellent biocompatibility with both cells. However, adding 5% DMSO (positive control) led to a cell viability of less than 50% for two cells (Figure 6A). Similarly, the cells cultured with AP-MN extract were all alive, and there was no significant difference from the normal medium group. At the same time, much necrosis occurred in the 5% DMSO group both in L929 (Figure 6B) and HaCaT cells (Figure 6C). Accordingly, it was concluded that AP-MNs were safe and biocompatible. 

### 3.5. Swelling Characterization of AP-MNs

As shown in Figure 7A, AP-MNs can rapidly swell to an equilibrium within 30 min after immersion in PBS buffer. Afterward, there were no apparent changes in swelling ratio with the extension of immersion time. Macroscopically, the diameter of AP-MNs was 0.8 cm before swelling and 1.2 cm after swelling for 24 h (Figure 7(C-1,C-2)). To observe the morphology changes of needles’ tips during the swelling process, 5% agar gel was applied to simulate the ex vivo skin. Before the experiment, AP-MNs had a sharp tip, and the microneedles’ height and base width were approximately 485 μm and 246 μm, respectively. After being inserted into the agar for 1 h, the volume of the AP-MNs rapidly expanded, manifested as the needles’ heights becoming shorter and the bases becoming thicker (Figure 7(C-3,C-4)). As the applied time increased, the height further shortened, and the substrate further widened (Figure 7(C-5,C-6)). The microneedles had swelled and balanced after 1 h (Figure 7B,C), indicating that the tips were no longer sharp, and the needles were completely swollen with water. Interestingly, they maintained a complete conical needle shape, consistent with the essential characteristics of HF-MNs, that is, swelling without dissolution. This property ensured that the AP-MNs could be removed entirely from the skin surface after completing drug delivery into the skin from a reservoir, reducing the possibility of topical accumulation of matrix materials in the skin.

### 3.6. In Vitro Transdermal Permeation Studies

Cumulative transdermal delivery curves of AZA and MAT are displayed in Figure 8. AZA and MAT could not effectively penetrate the skin within 10 h when the drug solution was directly dropped onto the surface of intact skin. Even if left on the skin for 24 h, the permeation ratio of both AZA and MAT was less than 5%. In contrast, the 24 h cumulative transdermal delivery ratios of AZA and MAT both exceeded 50% through AP-MN-assisted penetration. Additionally, it was found that the transdermal release of AZA and MAT remained consistent. In another study, it was reported that the permeability of AZA and MAT was inconsistent for dissolving microneedles loaded with AZA and MAT using sodium carboxymethyl cellulose (CMC) as the microneedles’ material. Specifically, AZA’s cumulative transdermal delivery ratio was always higher than that of MAT on account of the electronegativity of both CMC and AZA, as well as the electropositivity of MAT [33]. This discovery was consistent with the fact that the two materials used in the current microneedle formulation, PVA and Eudragit NM30D, were electrically neutral, and there was no interaction between the excipients and drugs. Thus, the simultaneous release of acidic and alkaline drugs can be achieved.

## 4. Conclusions

The AP-MNs composed of PVA and Eudragit NM30D were first developed based on a three-dimensional network structure formed by the physical entanglement of two materials. Through formulation optimization, the final composition of AP-MNs was determined to be 12% PVA + 6% Eudragit NM30D with the addition amount of MN solution of 30 μL per well. The microneedles possessed sharp tips, a complete array, and sufficient mechanical strength to penetrate the skin stratum corneum barrier effectively. FTIR and SEM studies confirmed the hydrogel network system, which was attributed to the physical blending and encapsulation effect between hydrophilic PVA and hydrophobic Eudragit NM30D after film formation. Laser confocal imaging indicated that applying AP-MNs improves the transdermal permeability of hydrophilic drugs. Further, the simultaneous releases of acidic and alkaline drugs were observed, suggesting MNs with electrically neutral materials did not affect the delivery of charged drugs. According to swelling characterization tests, the area of AP-MNs after complete swelling was approximately two times the initial area within 30 min, thus achieving a rapid drug delivery with a safe and friendly route. 

## Figures and Tables

**Figure 1 pharmaceutics-15-02007-f001:**
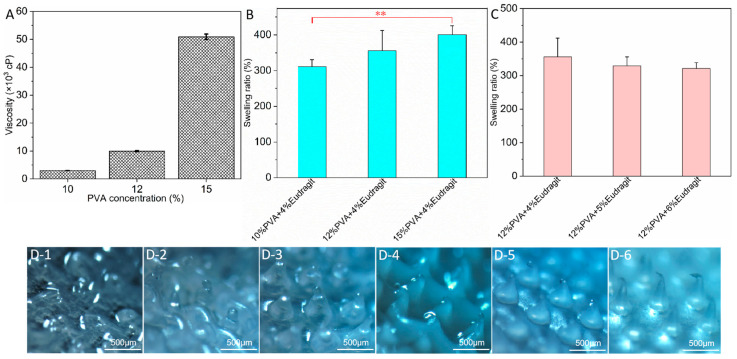
(**A**) Viscosity of PVA solutions with 10, 12, and 15% concentrations (mean ± SD, n = 3); (**B**) Swelling performance of microneedles composed of 10% PVA + 4% Eudragit NM30D, 12% PVA + 4% Eudragit NM30D, and 15% PVA + 4% Eudragit NM30D, respectively (mean ± SD, n = 3, ** represents *p* < 0.01); (**C**) Swelling performance of microneedles composed of 12% PVA + 4% Eudragit NM30D, 12% PVA + 5% Eudragit NM30D, and 12% PVA + 6% Eudragit NM30D, respectively (mean ± SD, n = 3); (**D**) The morphological changes of microneedles prepared from 12% PVA and different proportions of Eudragit NM30D (images **D1–6** were 0, 2, 3, 4, 5, 6%, respectively) after 8 h of application to the skin.

**Figure 2 pharmaceutics-15-02007-f002:**
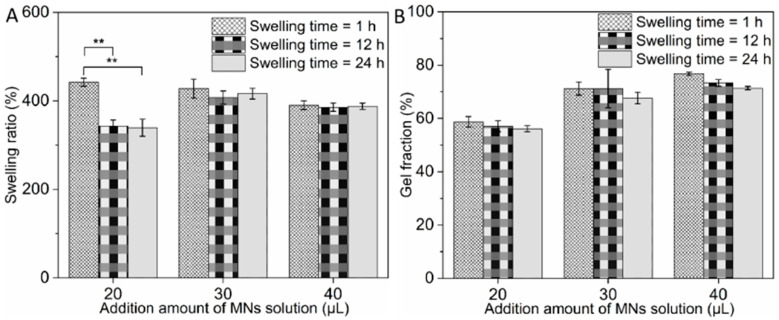
(**A**) Swelling ratios of microneedles composed of 12% PVA + 6% Eudragit NM30D with addition amounts of MN solution of 20, 30, and 40 μL, respectively, after immersion in PBS buffer for 1, 12 and 24 h (mean ± SD, n = 6, ** represents *p* < 0.01); (**B**) Gel fractions of microneedles composed of 12% PVA + 6% Eudragit NM30D with addition amounts of MN solution of 20, 30, and 40 μL, respectively, after immersion in PBS buffer for 1, 12 and 24 h (mean ± SD, n = 6).

**Figure 3 pharmaceutics-15-02007-f003:**
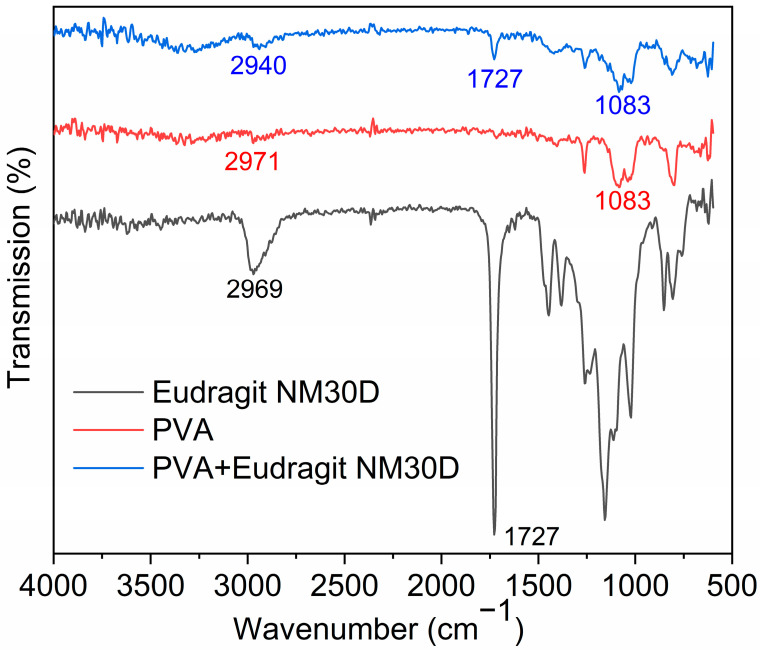
FTIR spectrograms of PVA, Eudragit NM30D, and PVA + Eudragit NM30D with wavenumbers from 600 to 4000 cm^−1^.

**Figure 4 pharmaceutics-15-02007-f004:**
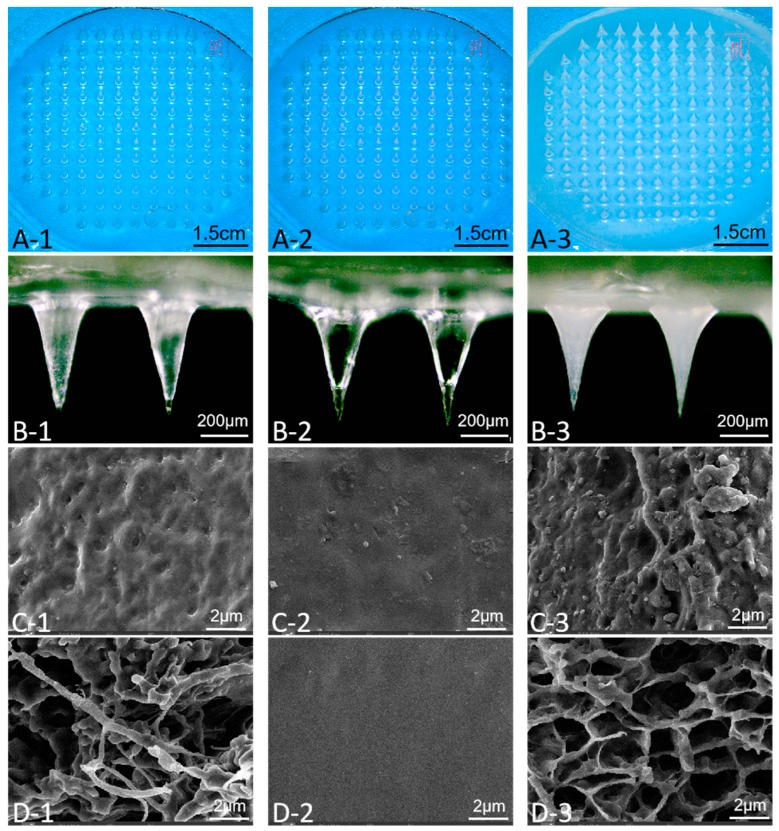
(**A**) Stereomicroscopy images of integral microneedles prepared with PVA, Eudragit NM30D, and PVA + Eudragit NM30D, respectively; (**B**) Bright field images of microneedles prepared with PVA, Eudragit NM30D, and PVA + Eudragit NM30D, respectively; (**C**) SEM images of microneedles prepared with PVA, Eudragit NM30D, and PVA + Eudragit NM30D, respectively; (**D**) SEM images of microneedles prepared with PVA, Eudragit NM30D, and PVA + Eudragit NM30D, respectively, after swelling in PBS buffer (0.01 M, pH = 7.4) for 24 h.

**Figure 5 pharmaceutics-15-02007-f005:**
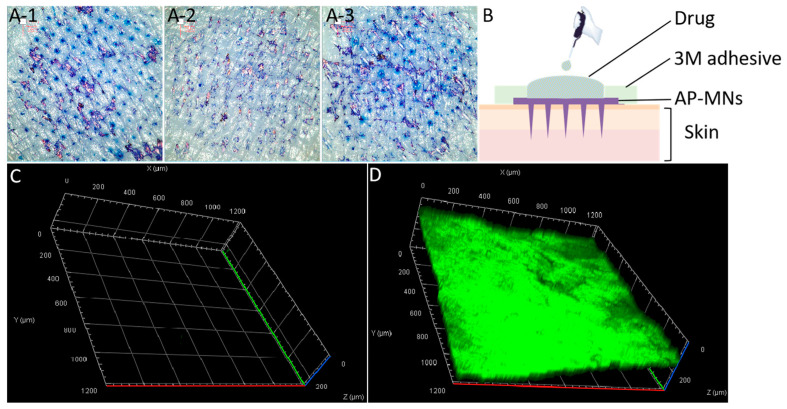
(**A**) Pinhole arrays on the porcine skin ex vivo after inserted with microneedles prepared with PVA, Eudragit NM30D, and PVA + Eudragit NM30D, respectively; (**B**) The schematic diagram of the transdermal delivery of drug under the assistance of AP-MNs; (**C**) Laser confocal fluorescence imaging of SF through free diffusion in the skin; (**D**) Laser confocal fluorescence imaging of SF through AP-MNs in the skin (Green fluorescence represents the penetration of SF into the skin).

**Figure 6 pharmaceutics-15-02007-f006:**
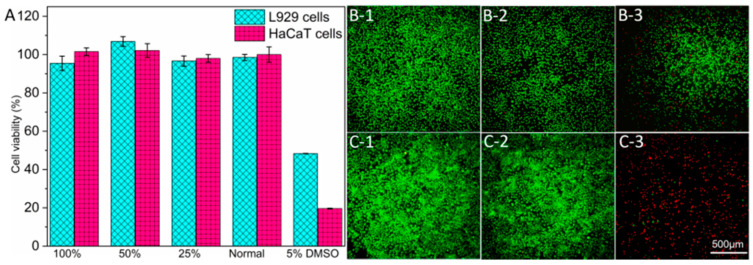
(**A**) Cell viability of the in vitro cytotoxicity study in L929 and HaCaT cells (mean ± SD, n = 4); (**B**) Dead/live states of L929 cells after incubation for 24 h (**1**: AP-MN extract; **2**: normal medium; **3**: 5% DMSO); (**C**) Dead/live states of HaCaT cells after incubation for 24 h (**1**: AP-MN extract; **2**: normal medium; **3**: 5% DMSO). Note: because the cell experiments of AP-MNs in this project were conducted simultaneously with the DMNs in previous studies [31], the results for the negative and positive groups were the same as those used in that published article.

**Figure 7 pharmaceutics-15-02007-f007:**
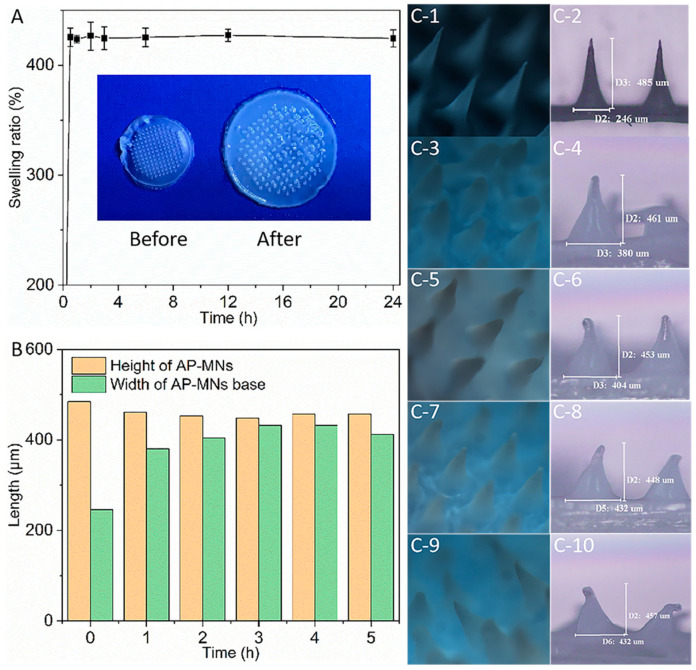
(**A**) Swelling curves of AP-MNs and physical appearances of AP-MNs before and after swelling (mean ± SD, n = 4); (**B**) Changes in needle height and width during the swelling process of AP-MNs on 5% agar gel; (**C**) Morphologies, height, and width of AP-MNs after swelling on 5% agar gel for 0, 1, 2, 3, 4 h, respectively (**C-1**, **C-3**, **C-5**, **C-7**, **C-9** were 3D microscope images of AP-MNs swelling for 0, 1, 2, 3, 4 h, respectively, while **C-2**, **C-4**, **C-6**, **C-8**, **C-10** were planar microscope images of AP-MNs swelling for 0, 1, 2, 3, 4 h, respectively).

**Figure 8 pharmaceutics-15-02007-f008:**
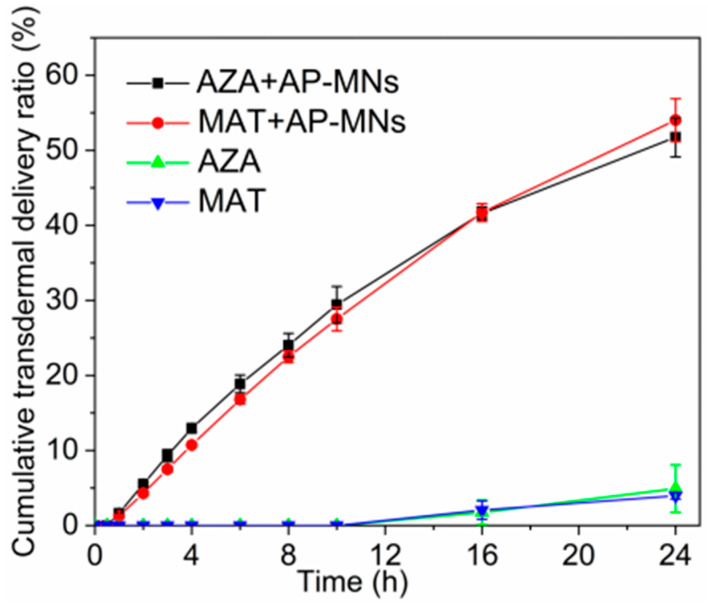
Cumulative transdermal delivery curves of AZA and MAT with and without the assistance of AP-MNs (mean ± SD, n = 3).

## Data Availability

Data available on request due to restrictions, e.g., privacy or ethical; the data presented in this study are available on request from the corresponding author.

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
