# Peer review of "Preparation and Evaluation of Auxiliary Permeable Microneedle Patch Composed of Polyvinyl Alcohol and Eudragit NM30D Aqueous Dispersion"

_pharmaceutics, 2023, doi:10.3390/pharmaceutics15072007_

Round 1
Reviewer 1 Report
Introduction
In our other study, we found that a porous network 100 structure was formed inside H-PVA MNs after absorbing phosphate (PBS) buffer, which 101 is suitable for promoting drug permeation channels (has submitted). What do authors mean by “has submitted”?
Please make the objective of the study clearer, indicating why the authors use Eudragit?
Authors may refer to the recent review on microneedles (Beneath the Skin: A Review of Current Trends and Future Prospects of Transdermal Drug Delivery Systems) Pharmaceutics, May 2022.
Methods
In Table S1: add 0% instead of (/).
Why the composition of microneedles mentioned in Figure 1C was not added to Table S1?
How did the authors measure the viscosity of the microneedle solution, why only one single reading?
What do you mean by “Security and compatibility” and why did you use two cell lines?
Results
Generally, a viscosity range of microneedles solution from 10000 268 to 20000 cp was preferred to prepare polymeric microneedles via the drop addition 269 method. Support by reference(s).
Figure 5, where is the legend of figure 5D?
Author Response
Response to Reviewer 1 Comments
Point 1: In our other study, we found that a porous network structure was formed inside H-PVA MNs after absorbing phosphate (PBS) buffer, which is suitable for promoting drug permeation channels (has submitted). What do authors mean by “has submitted”?
Response 1: What we want to express is a research article (AJOPS-D-23-00296) that has already been submitted to another journal. Sorry for the misunderstanding caused by our inappropriate expression, and it has been corrected in the revised manuscript.
Point 2: Please make the objective of the study clearer, indicating why the authors use Eudragit?
Response 2: Thank you for your kind suggestion. The Eudragit was used for preparation of AP-MNs on account of its safety, film-forming and water vapor barrier properties. The clearer description of the objective of the study has been supplemented in the revised manuscript.
Point 3: Authors may refer to the recent review on microneedles (Beneath the Skin: A Review of Current Trends and Future Prospects of Transdermal Drug Delivery Systems) Pharmaceutics, May 2022.
Response 3: We have supplemented and revised the references, and thank you for your advice.
Point 4: In Table S1: add 0% instead of (/).
Response 4: This error has been corrected.
Point 5: Why the composition of microneedles mentioned in Figure 1C was not added to Table S1?
Response 5: The formulation of MNs shown in Table S1 is a preliminary screening of matrix materials, and based on this, the ratio of PVA and Eudragit NM30D is further optimized. Therefore, the composition of microneedles mentioned in Figure 1C was not added to Table S1 to ensure the rationality of the logical order of the manuscript.
Point 6: How did the authors measure the viscosity of the microneedle solution, why only one single reading?
Response 6: The viscosity of the microneedle solution was measure with a viscometer (DV2T, BROOKFIELD, USA). In detail, a 50 mL centrifuge tube was used to prepare a 30 mL microneedle solution. After the solution was completely dissolved and the bubbles were removed by centrifugation, the centrifuge tube was placed in a 25 ℃ water bath for a constant temperature. Subsequently, the solution viscosity was recorded according to the viscometer. Furthermore, the results in Figure 1A were presented as mean ± SD (n =3). The error bar has been thickened to make it easy for readers to view.
Point 7: What do you mean by “Security and compatibility” and why did you use two cell lines?
Response 7: It refers to the cellular safety and biocompatibility of MNs. Thank you for the reminder. Our previous description was indeed inappropriate and has been corrected to “cell activity and biocompatibility”. As for the selection of cell lines, first, L929 cells were selected because they were the most commonly used in cytotoxicity tests. Besides, HaCaT cells belong to keratinocyte, and the AP-MNs were designed to skin application, which was suitable to using HaCaT cells.
Point 8: Generally, a viscosity range of microneedles solution from 10000 to 20000 cp was preferred to prepare polymeric microneedles via the drop addition method. Support by reference(s).
Response 8: We apologize for raising a viewpoint that has not yet been systematically validated. This viscosity range is limited to the empirical summary of the formulations used in this study. On the one hand, a low solution viscosity may cause the hollow state of microneedles, which leads to a decrease in the mechanical strength. On the other hand, when the solution viscosity is too high, the liquid injector cannot be used normally, resulting in inaccurate addition amount. In fact, there is currenty no systematic study on the viscosity of microneedle solution for polymeric microneedles prepared by molding method. Because there are various types of materials available for microneedles’ preparation, and each material has different polymerization degree, molecular weight, concentration, and other factors that lead to differences in solution viscosity and microneedle performance. As a fundamental research, we are also considering conducting a systematic review of this topic in future experiments and look forward to your continued attention. It was terribly sorry for the misunderstanding caused to you and the inappropriate statement has been corrected in the revised manuscript in red font.
Point 9: Figure 5, where is the legend of figure 5D?
Response 9: We are grateful for the kind reminder from the reviewer. The legend of Figure 5D has been added.

Reviewer 2 Report
Objectives of the study should be clearly mentioned in the introduction part.
Why the model drugs are the acidic and alkaline drugs? The background of these model drugs should be clearly mentioned in the introduction part.
Line 33, the ‘transdermal release” should be replaced with “permeation”.
Does “serial number” refer to “formulation code” ?
It is not clear how the AP-MNs form. Please clarify more in detail (2.2. Fabrication of AP-MNs).
Detail in “2.3. Screening of matrix composite ratio” should be rewritten and restructured.
In Table S1, what is “/” ?
Table and Figure number should be rearranged. There are Table S1 and Table 1, 2, 3 …, Figure S1, and Figure 1, 2, 3 appeared in the manuscript
.The clarity of Figure 1D should be improved.
The number of replications of each result in the Table/Figure is not stated. The statistically significant in some result was not clarified. The difference in Figure 1B and 1C seems to be insignificant.
Line 268-270: please clarify the scientific background of this statement.
It was not clear regarding the “addition amount of MNs solution”. This content should be rewritten and restructured.
More discussion that explores the relevant aspects of experiment result must be included.
Entire manuscript contains quite some spelling and grammatical mistakes. The whole manuscript is required to check and to correct these mistakes.
Author Response
Response to Reviewer 2 Comments
Point 1: Objectives of the study should be clearly mentioned in the introduction part.
Response 1: Thank you for your advice. The clearer objectives of the study has been supplemented in the revised manuscript.
Point 2: Why the model drugs are the acidic and alkaline drugs? The background of these model drugs should be clearly mentioned in the introduction part.
Response 2: According to your suggestion, we have supplemented the background of the two model drugs, as well as explained the reason why the model drugs are the acidic and alkaline drugs in the introduction part.
Point 3: Line 33, the ‘transdermal release” should be replaced with “permeation”.
Response 3: We have corrected the error according to your kind remainder.
Point 4: Does “serial number” refer to “formulation code” ?
Response 4: Yes, we are sorry for this ambiguous statement. It has been revised in red font.
Point 5: It is not clear how the AP-MNs form. Please clarify more in detail (2.2. Fabrication of AP-MNs).
Response 5: More detailed descriptions on the preparation of AP-MNs have been supplemented in the revised manuscript.
Point 6: Detail in “2.3. Screening of matrix composite ratio” should be rewritten and restructured.
Response 6: It has been rewritten and restructured in the revised manuscript in red font for your review again.
Point 7: In Table S1, what is “/” ?
Response 7: It should be replaced with “0%”, and this error has been corrected.
Point 8: Table and Figure number should be rearranged. There are Table S1 and Table 1, 2, 3 …, Figure S1, and Figure 1, 2, 3 appeared in the manuscript.
Response 8: According to the reviewer’ suggestions, the numbers of tables, figures, and supplementary materials have been rearranged to ensure the rationality and logicality of the article.
Point 9: The clarity of Figure 1D should be improved.
Response 9: The clarity of Figure 1D has been improved for readers to review.
Point 10: The number of replications of each result in the Table/Figure is not stated. The statistically significant in some result was not clarified. The difference in Figure 1B and 1C seems to be insignificant.
Response 10: The number of replications of each result has been added in the manuscript. In addition, the statistically significant in all data have been clarified to facilitate readers' understanding. Thank you for your meticulous review and advice.
Point 11: Line 268-270: please clarify the scientific background of this statement.
Response 11: We apologize for raising a viewpoint that has not yet been systematically validated. This viscosity range is limited to the empirical summary of the formulations used in this study. On the one hand, a low solution viscosity may cause the hollow state of microneedles, which leads to a decrease in the mechanical strength. On the other hand, when the solution viscosity is too high, the liquid injector cannot be used normally, resulting in inaccurate addition amount. In fact, there is currenty no systematic study on the viscosity of microneedle solution for polymeric microneedles prepared by molding method. Because there are various types of materials available for microneedles’ preparation, and each material has different polymerization degree, molecular weight, concentration, and other factors that lead to differences in solution viscosity and microneedle performance. As a fundamental research, we are also considering conducting a systematic review of this topic in future experiments and look forward to your continued attention. It was terribly sorry for the misunderstanding caused to you and the inappropriate statement has been corrected in the revised manuscript in red font.
Point 12: It was not clear regarding the “addition amount of MNs solution”. This content should be rewritten and restructured.
Response 12: It has been rewritten and restructured in the revised manuscript in red font for your review again.
Point 13: More discussion that explores the relevant aspects of experiment result must be included.
Response 13: As you mentioned, more discussion on experimental results has been supplemented and improved in the revised manuscript.
Point 14: Entire manuscript contains quite some spelling and grammatical mistakes. The whole manuscript is required to check and to correct these mistakes.
Response 14: We have carefully checked the language expression of the full text to standardize the scientific and rigor of the manuscript.

Reviewer 3 Report
The paper presents a good piece of work, the experimental design is good, and the paper can be published, after minor revision, concerning more the English and the editing of the manuscript, as detailed below.
- line 130: what is "human immortalized edipdermal"?
- in the text appear table S1 and figure S1 and S2. At the end it is clear these are for Supplementary material, but should be clearly stated in the text, and material removed and set in separate document.
- Figute S1. Not clear what should be seen, scale is missing, and looking at the caption, it seems id does not correspond to the order of figure. It seems to me that the only one conserving the morphology, if I interpreted correctly the figure, is the #1, that corresponds to 0% of Eudragit.
- line 317-8: please check significant digits!
- Figure 4: add visible bar scale in all images.
- Caption of Figure 5: it does not correspond to figures (B added but not described)
- Figure 6C not quoted in the paper.
General revision is suggested, some specific errors noted listed below:
- line 73: has widely used?
- line 102: what the authors means with "(as submitted)"? if it is a ref to a submitted paper, it is not admissible and is not clear.
- line 142-3 (revise)
- line 151: welling = swelling?
- line 259
- line 280-3
- caption Figure 2
- ref 6 (1008-+) and 12 (Ca2+)
Author Response
Response to Reviewer 3 Comments
Point 1: - line 130: what is "human immortalized edipdermal"?
Response 1: We sincerely apologize for the misspelling in the manuscript, which should be human immortalized keratinocytes (HaCaT). And it has been revised now.
Point 2: - in the text appear table S1 and figure S1 and S2. At the end it is clear these are for Supplementary material, but should be clearly stated in the text, and material removed and set in separate document.
Response 2: We agree with your comment and have removed table S1, figure S1, and figure S2 from the original manuscript and listed them separately in the supplementary materials.
Point 3: - Figute S1. Not clear what should be seen, scale is missing, and looking at the caption, it seems id does not correspond to the order of figure. It seems to me that the only one conserving the morphology, if I interpreted correctly the figure, is the #1, that corresponds to 0% of Eudragit.
Response 3: We have provided the more detailed captions, corresponding to the order of each figure to facilitate readers' understanding. Subsequently, the scale of Figure S1 has been added in the revised manuscript. Thank you for your serious and rigorous review.
Point 4: - line 317-8: please check significant digits!
Response 4: We are grateful for the kind reminder from the reviewer. It has been checked and confirmed.
Point 5: - Figure 4: add visible bar scale in all images.
Response 5: The visible bar scales have been added in each image.
Point 6: - Caption of Figure 5: it does not correspond to figures (B added but not described)
Response 6: Caption of Figure 5 has been added and corresponded to each image.
Point 7: - Figure 6C not quoted in the paper.
Response 7: Figure 6C has been quoted in the appropriate position in the paper.
Point 8: General revision is suggested, some specific errors noted listed below:
- line 73: has widely used?
- line 102: what the authors means with "(as submitted)"? if it is a ref to a submitted paper, it is not admissible and is not clear.
- line 142-3 (revise)
- line 151: welling = swelling?
- line 259
- line 280-3
- caption Figure 2
- ref 6 (1008-+) and 12 (Ca2+)
Response 8: Thanks a million for your valuable suggestions, the above mistakes have been carefully corrected and verified in the revised manuscript. Meanwhile, we have also carefully checked the language expression of the full text to standardize the scientific and rigor of the manuscript.

Round 2
Reviewer 1 Report
Authors respond to all comments.
Minor editing.